# *WSB1* Involvement in Prostate Cancer Progression

**DOI:** 10.3390/genes14081558

**Published:** 2023-07-29

**Authors:** Laura Boldrini, Massimo Bardi

**Affiliations:** 1Department of Surgical, Medical, Molecular Pathology and Critical Area, University of Pisa, 56126 Pisa, Italy; 2Department of Psychology & Behavioral Neuroscience, Randolph-Macon College, Ashland, VA 23005, USA

**Keywords:** prostate cancer, *WSB1*, survival, MDS

## Abstract

Prostate cancer (PC) is polygenic disease involving many genes, and more importantly a host of gene–gene interactions, including transcriptional factors. The *WSB1* gene is a transcriptional target of numerous oncoproteins, and its dysregulation can contribute to tumor progression by abnormal activation of targeted oncogenes. Using data from the Cancer Genome Atlas, we tested the possible involvement of *WSB1* in PC progression. A multi-dimensional scaling (MDS) model was applied to clarify the association of *WSB1* expression with other key genes, such as *c-myc*, *ERG*, *Enhancer of Zeste 1 and 2* (*EHZ1* and *EZH2*), *WNT10a*, and *WNT 10b*. An increased *WSB1* expression was associated with higher PC grades and with a worse prognosis. It was also positively related to *EZH1, EZH2*, *WNT10a*, and *WNT10b*. Moreover, MDS showed the central role of *WSB1* in influencing the other target genes by its central location on the map. Our study is the first to show a link between *WSB1* expression and other genes involved in PC progression, suggesting a novel role for *WSB1* in PC progression. This network between *WSB1* and *EZH2* through *WNT/β-catenin* may have an important role in PC progression, as suggested by the association between high *WSB1* expression and unfavorable prognosis in our analysis.

## 1. Introduction

Prostate cancer (PC) is the most common cancer in men in the western world and it is estimated that over 350,000 men worldwide die of PC every year, with the second-highest mortality among American men [1]. Although overall cancer mortality rates continue to decline, this progress may be attenuated by rising incidence for breast, prostate, and uterine corpus cancers, which also happen to have the largest racial disparities in mortality, considering that black men benefit more from screening in general and from the integration of personalized biomarkers because they are more likely to harbor genomically aggressive cancer, even with clinically low-risk disease [1]. Indeed, it was recently reported that, after two decades of decline, PC incidence is now increasing by 3% annually [1]. It is clear, therefore, that a great need remains to improve how PC is diagnosed and treated. Sub-stratification of PC into genetic subtypes can represent the basis of a more rational therapy for PC, especially considering that current clinical tools fail to reliably differentiate aggressive tumors from non-aggressive ones in order to predict therapeutic response [2].

PC is a polygenic disease involving many genes, and more importantly a host of gene–gene interactions [3]. Over two hundred susceptibility loci have been identified in genome-wide association studies [4], explaining why PC has one of the highest heritability scores of all cancers at 57% [5]. Nevertheless, polygenic risk scores have not been particularly effective in improving the ability to identify men with PC, providing accuracy comparable with the serum prostate-specific antigen (PSA) test and family history [6]. Gleason grading, introduced in the late 1960s, represented for several decades one of the most successful supports in clinical routine [7]. The most recent revision of the Gleason grading system, according to The International Society of Urological Pathology (ISUP) consensus recommendations and also adopted by the WHO classification of prostate [8,9,10], represented a significant improvement in managing prostate cancer patients. However, alongside the only grade assessment, the evaluation of new prognostically robust factors can provide a further refining in patients risk stratification [11]. For example, studies focusing on specific target genes showed a slightly better predictive outcome because selecting patients with a high grade of proliferation and DNA repair activity can improve an early identification of an aggressive PC with potentials for metastatic development [12].

A class of target genes with the potential to increase the early cancer identification and tumor development is constituted by those involved in DNA damage response by targeting homeodomain-interacting protein kinase 2, such as *WSB1* [13]. This protein (*WSB1*) is a member of the WD-protein subfamily, which contains seven WD40 repeats and a SOCS box in the C-terminus [14,15]. It is a direct transcriptional target of several other proteins involved in cancer metastasis [16]. An integrative genomics approach identifies *WSB1* as a target gene of Hypoxia Inducible Factor-1 (*HIF1*) and *c-myc* [17,18] in the core response to hypoxia. In these instances, *WSB1* was found to be significantly upregulated under hypoxic conditions in osteosarcoma cells [19]. Moreover, *WSB1* expression has been associated with tumor incidence and metastatic potential in pancreatic and hepatocellular cancer [13,20]. However, how *WSB1* may contribute to tumor initiation and progression is unknown; Kim et al. [21,22] showed that *WSB1* promotes tumor metastasis by inducing VHL degradation and acts as a tumor promoting factor by mediating ATM degradation and so overcoming the main barrier of tumor formation. As far as we know, no data exist regarding the role of *WSB1* in PC.

Early detection of localized PC remains a key clinical challenge considering that life expectancy of patients for men with localized prostate cancer is high, as 99% over ten years, while late stages diseases, with the onset of distant metastases, have a very short survival (only 30% at five years) [23,24]. Several multidisciplinary investigations are ongoing in PC research, in order to increase the knowledge of the molecular basis of this disease and of its progression, improving the identification of new prognostic factors. The *WSB1* gene is a transcriptional target of numerous onco-proteins and dysregulation of transcriptional factors can in turn contribute to tumor progression by abnormal activation of targeted oncogenes. Among the most promising, we assessed the role of *c-myc*, which is expressed at every stage of cancer development and is one of the most common upregulated genes in PC [25,26], and of the enhancer of Zeste (both *EZH1* and *EZH2*), the catalytic subunit of the Polycomb repressive complex 2 (PRC2), again a gene strongly upregulated in PC [27,28], and also involved in the *c-myc* regulation [29,30]. Alternatively, *WSB1* can promote *c-myc* expression through the *WNT/β-catenin* pathway, comprising *WNT* ligands, which are 350–400 amino acids long lipid-modified glycoproteins and *CTNNB1* gene-encoded multifunctional β-catenin proteins [31]. In canonical *WNT/β*-catenin signaling, the activation of *WNT* leads to translocation of β-catenin into the nucleus, which in turn acts as a co-activator of transcription factors [32]. Among the different physiological and pathological functions performed by *WNT/β*-catenin signaling, its role in the induction and progression of different types of cancers has been described [33]. WNT signaling activation via *CTNNB1* amplification are also frequent, occurring in approximately 10–30% of advanced prostate cancer cases [34,35,36].

Fusion of androgen-regulated transmembrane protease serine 2 (*TMPRSS2)* and ETS transcription factor, v-ets erythroblastosis virus E26 oncogene homolog (*ERG)* genes is a common somatic alteration in PC, resulting in overexpression of *ERG* oncogene, a member of the ETS transcription factor family [37,38]. However, the role of *ERG*-status molecular subtyping in prognosis of prostate cancer is under debate. Several studies concluded high *ERG* expression as a good prognostic marker [39,40,41], whereas other publications showed an inverse association [42,43].

Based on the results in literature regarding other cancer types, we tested the possible involvement of *WSB1* in PC progression. Specifically, we investigated the association of *WSB1* expression with other above reported key target genes (*c-myc*, *ERG*, *EHZ1*, *EZH2*, *WNT10a*, and *WNT 10b*) using data obtained from The Cancer Genome Atlas (TCGA, https://tcga-data.nci.nih.gov/tcga/ accessed on 2 May 2023). Moreover, we used multidimensional scaling models (MDSs) in order to propose novel mechanisms of *WSB1*-mediated PC progression. MDS is a graphical method by which a set of variables or items (genetic associations in our sample) is represented by a set of points in two or higher dimensional space. As the latter often resembles a geographical map, MDS results are often referred as a ‘map’. The distance between two points on the MDS map is defined by the ‘dissimilarity’ of the corresponding variables. Although there is a multitude of possible dissimilarity measures that can be used in MDS, in our sample it was based on the Pearson correlation coefficient. The higher the correlation between the expression of two genes, the closer to each other should these items be represented on the MDS map. The aim of MDS was to represent these dissimilarities as accurately as possible by Euclidean distances in a low dimensional space, simultaneously for all the genes considered.

## 2. Materials and Methods

### 2.1. TCGA Database

We downloaded and processed the clinical data, such as age, radical prostatectomy grading, disease-free interval (DFI), overall survival (OS), and IlluminaHiSeq gene expression profiles of 496 PC patients from the TCGA data portal. The International Society of Urological Pathology (ISUP) consensus recommendations also adopted by the WHO classification of prostate [8,9,10] was used to update the Gleason score (GS) of the original TCGA data: Group 1 for GS ≤ 6; 2 for GS 7, 3 + 4; 3 for GS 7, 4 + 3; 4 for GS 8, and 5 for GS 9, 10. We decided to create two categories, including ISUP 1, 2, and 3 grades into low, and 4 and 5 grades into high group, in order to minimize the probability for too much variation. The age at the diagnosis was provided in years, whereas DFI and OS were expressed in months.

### 2.2. Statistical Analysis

Pearson’s r was used to calculate the bivariate correlations among clinical output and target gene expression, and to evaluate the correlation between *WSB1* expression and the other target genes. To test the differences in the average of gene expressions by GS, we used *t*-tests. A stepwise regression analysis was run to identify the best predictors of *WSB1* expression among the correlated genes. All analyses were considered significant at the α-level = 0.05

Survival analyses were performed using the Kaplan–Meier method with the Wilcoxon log-rank test for statistical significance. Kaplan–Meier is a non-parametric statistic used to estimate the survival function from lifetime data. In medical research, it is often used to measure the fraction of patients living for a certain amount of time after treatment. A plot of the Kaplan–Meier estimator is a series of declining horizontal steps which, with a large enough sample size, approaches the true survival function for that population. The value of the survival function between successive distinct sampled observations (“clicks”) is assumed to be constant. A typical application might involve grouping patients into categories, for instance, those with WBS1 low-activation and those with WBS1 high-activation. In the graph, the difference in DFI and OS can be estimated. To generate a Kaplan–Meier estimator, at least two pieces of data are required for each patient (or each subject): the status at last observation (event occurrence or right-censored), and the time to event (or time to censoring). If the survival functions between two or more groups are to be compared, then a third piece of data is required: the group assignment of each subject.

Finally, to identify the independent association among gene expressions, a Multi-Dimensional Scaling analysis was run. This technique is useful in mapping the similarities among the various genes. Distances between variables were derived looking at partial correlations (i.e., proximities) among variables, which were subsequently used to create a matrix of distance can be displayed graphically. The closer two or more variables are on the map, the more highly correlated they are, while the farther apart they are, the less correlated they are. To test the reliability of the model, we check for both S-stress (how well the variables fit into the model) and the percentage of the original variance explained (RSQ). Typically, S-stress of 0.1 or lower and RSQ of 0.8 or higher are considered acceptable.

All analyses were performed using SPSS 28.1 (IBM, Armonk, NY, USA).

## 3. Results

### 3.1. TCGA Database

The average age of the 496 PC ± patients was 61.03 ± 6.82 years (range 41–78 years). The reported DFI was 32.18 ± 24.84 months (range 1–165), OS was 35.78 ± 25.86 months (range 1–165). The age of diagnosis was negatively related to both DFI (r = −0.128; *p* = 0.004) and OS (r = −0.113; *p* = 0.012), whereas OS was positively related to the DFI (r = 0.914; *p <* 0.001).

Table 1 showed the distribution of the PC grading groups by ISUP consensus recommendations. Tumors with high grade, corresponding to groups 4 and 5, showed a higher probability to have a shorter DFI, with a significant *p*-value (t_488_ = 3.69; *p* = 0.05), confirming that men with high grade tumors had higher probability to have a shorter time without the disease (Figure 1). OS was not significant related to the grading groups.

*T*-test was used to check if different grading groups were related to the expression of the target genes; tumors with advanced grades showed higher *WSB1*, *EZH2*, and *WNT10b* expression levels (t_494_ = 6.005 *p* < 0.0001); t_494_ = 7.308 *p* < 0.0001; t_494_ = 1.98 *p* = 0.048, respectively), whereas the other genes were not significantly related to grade score (all *p*-values > 0.075).

### 3.2. Association among Gene Expressions

Among the target genes under investigation, it was found that gene expressions were highly correlated (Table 2). Specifically, *WSB1* was positively related to *EZH1*, *EZH2*, *WNT10a*, and *WNT10b*, but not to *ERG*, *Myc*, *HIF1alpha*, and *WSB2*.

To identify the independent association among the significant correlations an MDS was run (Figure 2). The S-stress was 0.061 and the RSQ was 99%, thus indicating a good reliability of this model. The central role of *WSB1* in influencing the other target genes was demonstrated by its central location on the map.

Finally, a stepwise linear regression analysis among the variables included in the MDS confirmed that the expression of *WSB1* was specifically associated with *EZH1* (step 1, R^2^ = 0.1), *EZH2* (step 2, R^2^ = 0.15), whereas adding both *WNT10b* (step 3, R^2^ = 0.15) and *HIF1alpha* (step 4, R^2^ = 0.15) did not significantly increase the predictive value of the model (Table 3).

### 3.3. Survival Curves

Kaplan–Meier curves showed a shorter DFI and OS in patients with high *WSB1* expression in comparison with patients with low *WSB1* levels (Generalized Wilcoxon χ^2^ = 13.280, *p <* 0.001 and χ2 = 6.196, *p* = 0.013, respectively) (Figure 3). Taken together with the previously reported association between elevated *WSB1* expression and high tumor grade, these survival data demonstrated that *WSB1* overexpression was correlated with unfavorable prognosis, suggesting *WSB1* as a clinical biomarker for prostate cancer.

## 4. Discussion

Currently, clinical factors are most used for prostate cancer patient’s prognosis evaluation. Introducing clinical tools such as the GS coupled with the more recent advances in PC grading have significantly optimized prognostic evaluation in clinical practice [7,8]. The usefulness of PC grading was confirmed in our study, where PC patients with high grade scores showed a significantly lower DFI. It was puzzling that in our dataset, OS was not significantly related to the grading groups. This result, which is in disagreement with most of the data in literature [10], can be interpreted in several ways. Very likely, this is due to the fact that patients included in our sample had a significantly lower than average level of cancer aggressivity, as confirmed by the high survival rates recorded in the database. This sampling problem could have created an unbalanced number of prostate cancer patients with low grade in our series, as also recently reported by Brundage et al. [44], who also reported that GS was not associated with either progression-free survival or overall survival in their cohort, a fact likely due to the infrequent number of patients in their study with a low or intermediate GS. Inconsistencies of this nature make even more relevant the need for molecular tools to be associated with the diagnosis and prognosis of PC. Predicting disease progression and prognosis plays an essential role in guiding patient goals, expectations, and treatment strategies. As the cost of genetic analysis decreases, it is becoming increasingly feasible for patients to undergo genetic and genomic screening to define and individualize the biology of each patient’s cancer. Ideal future models will likely incorporate some of these clinical factors in conjunction with direct measures of biology such as *WBS1* expression among other genomic alterations. Moreover, the benefit of testing not only one driver gene, but more probably a combination of factors is becoming evident, and in this way our new insights into a putative transcriptional regulatory circuit, controlling other gene expressions, may represent a tool for a better manage in prostate cancer patients.

This is why the main goal of the present study was to assess the potential of *WSB1* as an oncogene in PC. Our analysis confirmed this hypothesis, showing a link between *WSB1* expression and the progression of PC, in terms of both higher grades and clinical output (OS and DFI). It is of particular interest that results including both clinical data and *WSB1* expression were useful in discriminating OS in our database, whereas the latter by themselves were not. It was also hypothesized that *WSB1* can represent a key pathway in the activation of several other oncogenes. As a matter of fact, our results suggested this direct relationship, as indicated by the central role of *WSB1* in the MDS analysis.

*WSB1* is a member of the WD-protein subfamily, and it is a transcriptional factor as a part of an E3 ubiquitin ligase complex. Over the past two decades, the number of studies on E3 ubiquitin ligase and transcriptional factors has explosively grown, considering their role in cancer and other diseases [45]. Several studies have demonstrated that dysregulation of transcriptional factors is involved in carcinogenesis and tumor progression, and new insights into transcriptional programs dysregulation may represent a benefit in cancer patients management [46]. Vulnerabilities in transcriptional program can be predicted by genetic changes with consequent dysregulation of specific gene expression profiling and different transcriptional addiction seemed to be active in specific subsets of cancer. Alteration of transcriptional network, following transcription factors deregulation, is involved also in the prostate cancer model [47]. Considering that cancer arises from an interplay of different oncogenic hits, we made efforts trying to define a model of a transcriptional regulatory circuit, controlling other gene expression thereby driving prostate tumor behavior and progression.

*WSB1* is a *HIF*-target and seems to accelerate malignant progression in several tumors by increasing hypoxic microenvironment via *HIF1alpha* [19]; however, Haque et al. showed that [46] *WSB1* expression does not significantly increase *HIF1alpha* mRNA levels, as might be expected. Accordingly with this previous study, we did not find an association between *WSB1* and *HIF1alpha* gene expression level; the loop may have formed by *WSB1* and *HIF1alpha* can be also related to decreased protein degradation beyond through increased transcription, or to the degradation of diverse proteins associated with the cellular response to hypoxia. In fact, regulation of cellular response to hypoxia has been shown as only one of the key regulator functions of *WSB1* and other protein targets of *WSB1* are also involved in other cancer pathways [48].

The first novelty of our study was the prognostic value of *WSB1* in PC. The statistical difference in the Kaplan–Meier curves showed an involvement of *WSB1* in PC progression, as testified by patients with a higher radical prostatectomy grade were also characterized by a lower DFI. Although significantly different, in our data the curves were fairly close together, confirming that *WSB1* can be just one of many genetical checkpoints involved. Looking at the two curves more closely, we noticed that the biggest differences were for men with DFI between 10 and 50 months. At higher DFI levels, the two curves tended to overlap. This can be explained in many ways; we believe that because our database was mostly comprised by men with mid-grade tumors, the variability at higher DFI levels was artificially low. It would be of extreme interest to know the relationship between mRNA and protein expression. Unfortunately, the data available for us did not offer such correlation. This is a limitation of our report and certainly a future goal to assess the role of *WSB1* in PC progression and its potential use in clinical settings.

Also, as far as we know, our study is the first to show a link between *WSB1* expression and others target genes involved in PC progression, also suggesting a novel role for *WSB1* in PC progression and a key pathway downstream of *WSB1*. Based on our results of PC data extracted from the TCGA database, we believe that *WSB1* can recruit *EZH2* to the β-catenin transcriptional complex. *WNT/β*-catenin signaling comprises WNT ligands, which are 350–400 amino acids long, lipid-modified glycoproteins, and *CTNNB1* gene-encoded multifunctional β-catenin proteins [31]. In canonical *WNT/β*-catenin signaling, the activation of WNT leads to translocation of β-catenin into the nucleus, which in turn acts as a co-activator of transcription factors [49]; the stabilization of β-catenin protein is a key process in transducing WNT signaling, and this association specifically enhances *WNT* target gene transactivation. An up-regulation of the components of *WNT/β*-catenin, in particular of *WNT10a* and *WNT10b*, along with change in the expression pattern of *β-catenin* has been documented in cancer patients [33]. The central role of *WSB1* in our MDS analysis prompted us to think that *WSB1* mediates *EZH2*-induced WNT signaling hyperactivation. The body of evidence showed that *EZH2* was able to control several specific transcription factors [50]. One of the novelties of our study was the putative crosstalk between *WSB1* and *EZH2*, presenting important hints for future investigation of PC, in consideration of their association with advanced grades and their role as unfavorable prognostic factors. The different roles of the paralogues *EZH1* and *EZH2* need to be further studied in PC; data in literature showed a different expression pattern for *EZH1*, found in dividing and differentiated cells, and *EZH2*, only present in highly proliferative cells [51,52] in T-cell lymphomas patients, but no data exist in prostate cancer. There are also two WSB proteins, *WSB1* and *WSB2*, based on their number of WD motifs, with an amino acid sequence similarity of 65% and with surprisingly both similar and conflicting properties; more attention has been focused in literature on the function of *WSB1*, with dysfunctional *WSB1* expression closely related to tumorigenesis and progression [53]. Zhang et al. showed [54] high *WSB2* levels associated with clinicopathological features in patients with melanoma. In our model no correlation was found between *WSB1* and *WSB2*, confirming their different role in cell signal transduction in various cancer models.

*C-myc* is demonstrated to be involved not only in tumor initiation, but also in cancer progression [55], an important point to keep in mind considering that *WSB1* was also reported as a direct target gene of c-myc [56]. In our study we did not find a direct correlation of *WSB1* with *c-myc*; c-myc oncogene expression was correlated with EZH and WNT, so *WSB1* seemed to promote the expression of *c-myc*, not directly, but through the *WNT/β*-catenin pathway. This putative network between *WSB1* and *EZH2*, with *c-myc* activation through *WNT/β*-catenin may have an important role in PC progression, as suggested by the association between high *WSB1* expression and the unfavorable prognosis we found in our analysis.

The *TMPRSS2-ERG* gene fusion occurs in ~50% of PC patients [37], which makes it the most frequent alteration observed in human PC. Zoma et al. [57] recently reported that *EZH2* interacts with ERG, methylating it at lysine 362, and so increasing ERG oncogenic activity by enhanced transcription of ERG target genes. In our model, aberrant overexpression of ERG seemed to be linked to PC progression, as previously reported [58,59], by a transcriptional network with c-myc, *EZH1*, *WSB2*, and *WNTa* and *WNTb*, suggesting another way of *WNT/β*-catenin pathway activation.

In conclusion, we provided new insights in PC progression showing promises in targeting the transcription factor *WSB1*; however, working with the TGCA database has clear limitations that are shared in our work and the translation into patients’ management provides a clinical impact. Moreover, although the central role of *WSB1* in mediating other oncogenes appeared clear, alternative models involving different pathways of activations should be tested. Further investigation of this *WSB1*-mediated circuit will help to further clarify the PC biology, and potentially discover new potential therapeutic targets for the more aggressive PC forms.

## Figures and Tables

**Figure 1 genes-14-01558-f001:**
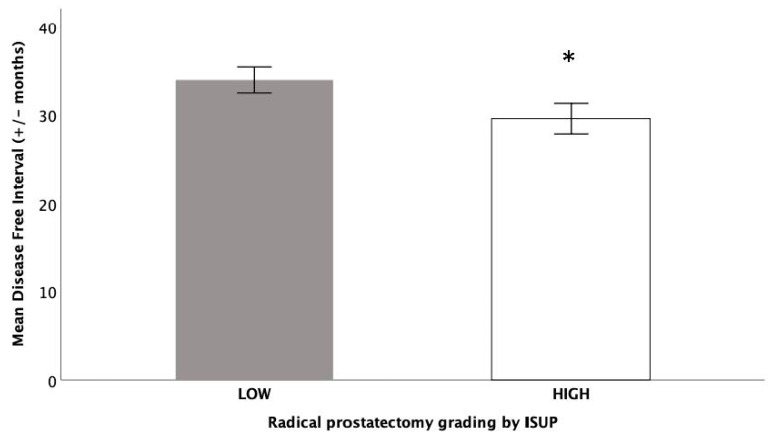
Differences in DFI among grading groups by the ISUP consensus recommendations. Low group included grade 1, 2, and 3 by ISUP (1 for GS ≤ 6; 2 for GS 7, 3 + 4; 3 for GS 7, 4 + 3); high group included grade 4, and 5 by ISUP (4 for GS 8, 4 + 4, 3 + 5, 5 + 3; 5 for GS 9 and 10, 4 + 5, 5 + 4, 5 + 5). * High grade PC showed a higher probability to have a shorter DFI, with a significant *p*-value (*p* = 0.05).

**Figure 2 genes-14-01558-f002:**
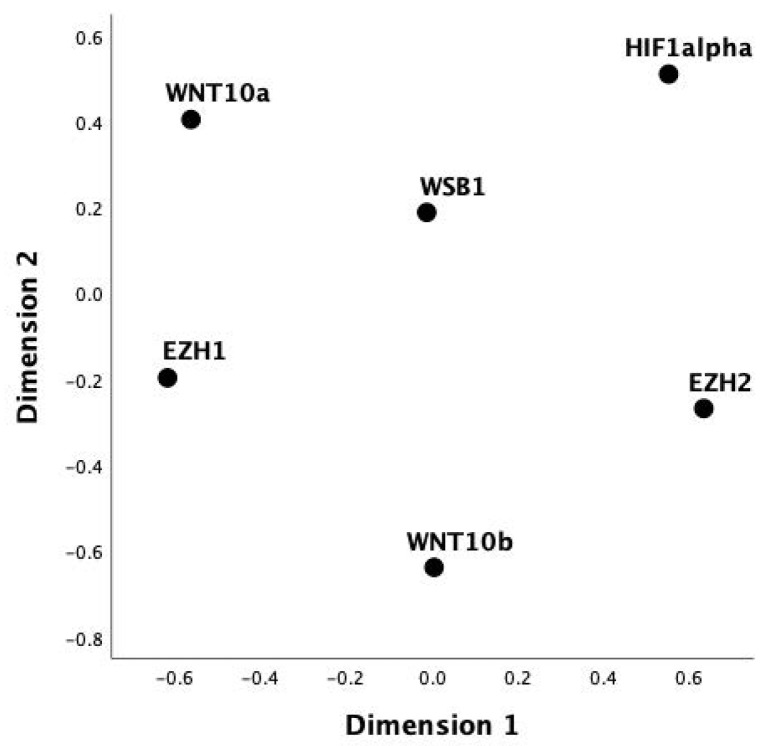
Map of the association among the variables included in the multidimensional scaling (MDS) model: the central role of *WSB1* in influencing the other target genes was demonstrated by its central location on the map.

**Figure 3 genes-14-01558-f003:**
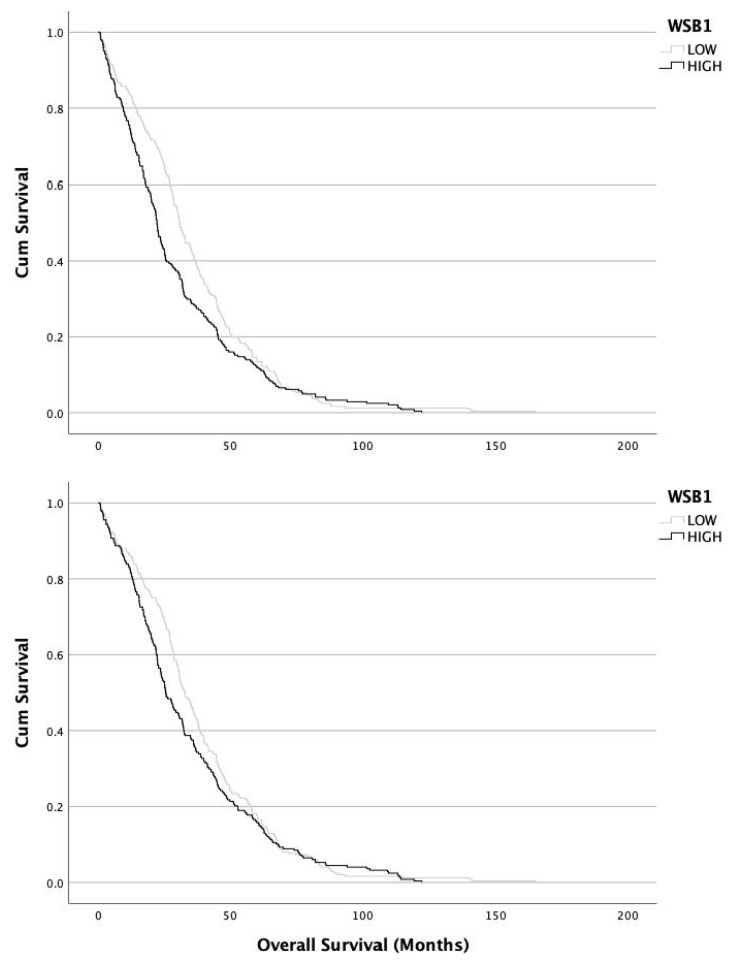
Kaplan–Meier curves showed a shorter DFI and OS in patients with high *WSB1* expression in comparison with patients with low *WSB1* levels (Generalized Wilcoxon χ^2^ = 13.280, *p <* 0.001 and χ^2^ = 6.196, *p* = 0.013, respectively).

**Table 1 genes-14-01558-t001:** Number of PC patients by ISUP consensus grading after radical prostatectomy.

Grade Group	GS	N
1	6 (3 + 3)	44
2	7 (3 + 4)	146
3	7 (4 + 3)	101
4	8 (4 + 4, 3 + 5, 5 + 3)	64
5	9, 10 (4 + 5, 5 + 4, 5 + 5)	141

**Table 2 genes-14-01558-t002:** Correlation among gene expressions.

	ERG	MYC	HIF1a	EZH1	EZH2	WSB2	WNT10b	WNT10A
WSB1	0.070	0.048	0.017	0.317 **	0.144 **	–0.070	0.106 *	0.129 **
ERG		0.113 *	0.019	–0.226 **	–0.039	0.458 **	0.144 **	–0.186 **
MYC			0.189 **	0.019	0.347 **	–0.005	0.131 **	–0.167 **
HIF1alpha				–0.250 **	0.130 **	0.201 **	–0.030	–0.219 **
EZH1					–0.222 **	–0.441 **	0.008	0.412 **
EZH2						0.113 *	0.050	–0.181 **
WNT10b							0.149 **	–0.243 **
WNT10a								0.037

(**) = Significant values *p* < 0.01 (two tails). (*) = Significant values *p* < 0.05 (two tails).

**Table 3 genes-14-01558-t003:** Stepwise regression of the dependent variable WS1 by other genes expression.

Model		Not-Standardized Coefficients		Standardized Coefficients	t	*p*
		B	Standard Error	β		
1	(Constant)	6221 K	1339 K		4.645	0.000
	*EZH1*	1.195	0.161	0.317	7.420	0.000
2	(Constant)	1853 K	1543 K		1.201	0.230
	*EZH1*	1.384	0.161	0.367	8.607	0.000
	*EZH2*	2.110	0.398	0.226	5.294	0.000
3	(Constant)	1258 K	1560 K		0.807	0.420
	*EZH1*	1.377	0.160	0.365	8.596	0.000
	*EZH2*	2.063	0.397	0.221	5.190	0.000
	*WNT10b*	4.104	1.845	0.092	2.224	0.027
4	(Constant)	−804 K	1851 K		−0.435	0.664
	*EZH1*	1.454	0.164	0.385	8.865	0.000
	*EZH2*	1.997	0.397	0.214	5.024	0.000
	*WNT10b*	4.229	1.840	0.095	2.298	0.022
	*HIF1-α*	0.058	0.028	0.088	2.055	0.040
**Model**	**R**		**R-Squared**	**Adapted** **R-Squared**	**Standard Error Value**	
1	0.317 ^a^		0.100	0.098	7,746,629.967	
2	0.386 ^b^		0.149	0.145	7,543,005.092	
3	0.396 ^c^		0.157	0.152	7,512,987.926	
4	0.405 ^d^		0.164	0.158	7,488,512.108	

^a^ Predictors: (constant), *EZH1*; ^b^ Predictors: (constant), *EZH1*, *EZH2*; ^c^ Predictors: (constant), *EZH1*, *EZH2*, *WNT10b*; ^d^ Predictors: (constant), *EZH1*, *EZH2*, *WNT10b*, *HIF1-α*.

## Data Availability

Not applicable.

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
