# Peer review of "WSB1 Involvement in Prostate Cancer Progression"

_genes, 2023, doi:10.3390/genes14081558_

Round 1
Reviewer 1 Report
The paper is interesting and well written. The methodology is well described and the discussion that tries to bridge the informatics findings with the biological functioning of the various proteins is interesting. A couple of concerns include:
1. Gleason grade was developed and utilized as a prognostic indicator in prostate cancer that should be associated with overall survival and/or disease free interval. The authors do not find this association, even when grouping the Gleason scores. Why not? This finding may bring some doubt into the fidelity of the study data.
2. WSB1 expression correlated with other negative features of prostate cancer (lines 159-161) with the authors suggesting it as a clinical biomarker. Why? What about the negative features that correlate with it? There are already a lot of molecular prognostic indicators/possibilities in prostate cancer. Do we need more? What value does WSB1 expression in particular offer? Does the mRNA match the protein expression?
3. The Kaplan-Meier curves are fairly close together (Fig 3). While the statistical difference is clear, is this difference actually clinically useful?
4. Studying mRNA expression profiles is not the same as studying protein levels. This manuscript does not show data concerning WSB1 protein expression (lines 168-171). Maybe there will be a correlation between the protein expression co-association data and maybe not. The authors should not confuse protein expression with mRNA expression in their discussion. There has not been a study of protein expression in this manuscript.
5. Minor grammatical flaws remain (eg. line 213, lines 242-244), though in general, the overall writing quality is good.
1. Minor grammatical flaws remain that can be improved with another round of editing.
Author Response
See file

Reviewer 2 Report
Abstract - no issues
Introduction - nicely written paragraph, clearly presenting the background and rationale for the study
Material and Methods - raw 91 - why not ISUP instead of GS? sub-grouping of Gleason 7?
Table 1 - percentage? looks like a typo
Results - nicely written paragraph - clearly presenting authors results on GS, DFI and OS with WSB1, as well as MDS model
Discussion - Authors` results are elegantly correlated with the literature, emphasizing the central role of WSB1 in prostate carcinogenesis
it is this reviewer`s opinion that the manuscript should be considered for publication after authors sufficiently addressed the aforementioned issues
Author Response
See file

Round 2
Reviewer 2 Report
the authors have answered sufficiently to the questions raised.
it is this reviewer recommendation to accept the manuscript for publication in the present form